# Immunogenicity of a Candidate DTacP-sIPV Combined Vaccine and Its Protection Efficacy against Pertussis in a Rhesus Macaque Model

**DOI:** 10.3390/vaccines10010047

**Published:** 2021-12-30

**Authors:** Xiaoyu Wang, Na Gao, Jiana Wen, Jingyan Li, Yan Ma, Mingbo Sun, Jiangli Liang, Li Shi

**Affiliations:** 1Laboratory of Vaccine Development, Institute of Medical Biology, Chinese Academy of Medical Science & Peking Union Medical College, Kunming 650118, China; gengshuai_wc@imbcams.com.cn (X.W.); wenjiana@imbcams.com.cn (J.W.); mayan2519@imbcams.com.cn (Y.M.); smb@imbcams.com.cn (M.S.); 2Key Laboratory of Vaccine Research and Development on Severe Infectious Diseases, Institute of Medical Biology, Chinese Academy of Medical Science & Peking Union Medical College, Kunming 650118, China; gaona@imbcams.com.cn (N.G.); lijingyan@imbcams.com.cn (J.L.); 3Laboratory of Immunogenetics, Institute of Medical Biology, Chinese Academy of Medical Science & Peking Union Medical College, Kunming 650118, China

**Keywords:** DTacP-sIPV candidate vaccine, immunogenicity, efficacy, rhesus macaque model

## Abstract

The research and development of a pertussis-combined vaccine using a novel inactivated poliovirus vaccine made from the Sabin strain (sIPV) is of great significance in the polio eradication project and to address the recent resurge in pertussis. In the present study, we compared the immunogenicity and efficacy of a candidate DTacP-sIPV with those of a commercial DTacP-wIPV/Hib, DTaP/Hib, pertussis vaccine, and aluminum hydroxide adjuvant control in the rhesus macaque model with a 0-, 1-, and 2-month immunization schedule. At day 28 after the third dose, rhesus macaques were challenged with aerosol pertussis and the antibody and cellular response together with pertussis clinical symptoms were determined. The production of anti-PT, anti-PRN, anti-FHA, anti-DT, anti-TT, and polio type I, II, III antibodies was induced by the candidate DTacP-sIPV, which was as potent as commercial vaccines. In comparison with the control group that showed typical pertussis symptoms of humans after the aerosol challenge, the DTacP-sIPV group did not exhibit obvious clinical pertussis symptoms and had higher neutralization titers of anti-PT, anti-PRN, and anti-FHA. In conclusion, the DTacP-sIPV vaccine was able to induce immunity in rhesus macaques to prevent pertussis infections after immunization. The developed vaccine was as efficient as other commercial vaccines.

## 1. Introduction

Combination vaccines are currently recommended for use in routine immunization schedules, owing to advantages such as the reduced number of injections, decreased discomfort for children, improved vaccine coverage rates, and cost effectiveness [1,2]. The diphtheria-tetanus-pertussis-inactivated poliovirus vaccine (DTP-IPV) has been used as one of the most effective combination vaccines in developed countries for several years [3,4,5]. However, in China and other developing countries, an oral poliovirus vaccine (OPV) based on attenuated Sabin strains was included in routine immunization for polio control and eradiation for almost 50 years because of its low cost, ease of administration, and good immunogenicity; hence, DTP and OPV are separately administrated [6]. With the progress in polio eradication, the World Health Organization (WHO) recommends the development of an inactivated poliomyelitis vaccine derived from an attenuated poliovirus Sabin strain (sIPV) that can significantly reduce the potential risk of outbreaks caused by containment breaches during vaccine production as compared with wild-type poliovirus strain-based IPV (wIPV), especially in developing countries. Both the conventional IPV made from wild poliovirus and the recently launched sIPV are included in the “Global Polio Eradication & Endgame Strategic Plan” [7]. Thus, the use of sIPV to develop a DTP-sIPV combined vaccine will facilitate polio eradication in China and other countries worldwide.

Immunization with DTP and DTP-based combined vaccines has managed to control the related infectious diseases. However, there has been a worldwide increase in the incidence of pertussis in recent years [8]. Even developed countries with high vaccination coverage have experienced a rise in the incidence of pertussis [9,10]. Pertussis resurgence is reported not only in infants and toddlers under 5 years of age but also in adolescents (14–16 years old) and elderly people over 65 years of age [11,12]. The re-emergence of pertussis may be caused by genetics changes in Bordetella pertussis, reduced vaccine efficacy, the rapid occurrence of waning immunity, increased recognition and reporting of pertussis, and laboratory diagnosis movement. The reduced vaccine efficacy may be related to the acellular pertussis (aP) vaccine, which has replaced the whole-cell pertussis (wP) vaccine mostly to reduce side-effects [13,14]. Studies in a baboon model of pertussis have suggested that the wP vaccine can prevent both the disease and viral transmission, while the aP vaccine can only protect the recipient from the disease [15]. In our previous study, we challenged the infant rhesus macaque with B. pertussis using an aerosol method and demonstrated its applicability as another non-human primate model for B. pertussis for new pertussis-related vaccine studies [16].

Meanwhile, we developed DTaP using a co-purification process for the pertussis vaccine, and confirmed its good immunogenic effects in rhesus macaques [17]. In the present study, we improved the purification process of the pertussis component and developed a three-component aP formulation to manufacture a combined DTacP-sIPV vaccine. Then, we evaluated the immunogenicity and protective effects of the DTacP-sIPV vaccine in a rhesus pertussis model to select a suitable candidate formulation for clinic trials. 

## 2. Materials and Methods

### 2.1. Animal Use and Welfare

According to our previous successful pertussis model on infant rhesus macaques, which showed typical whooping cough, leukocytosis, bacteria-positive nasopharyngeal wash, inter-animal transmission, and the systemic immune response after pertussis infection, which matched human children infected with pertussis [17], twenty-nine healthy rhesus macaques, half male and half female, aged 3–4 months (12–15 weeks) were obtained from the Institute of Medical Biology, Chinese Academy of Medical Sciences (IMBCAMS) (Animal License No. SCXK (Dian) K2015-0004). Rhesus macaques were housed in an animal house (room temperature 22–25 °C, relative humidity 40–60%, light and darkness each for 12 h) with free access to food and water. All animals were under the care of veterinarians at the Institute of Medical Biology, Chinese Academy of Medicine Sciences. The housing conditions, experimental procedures, and animal welfare were in accordance with the local laws and guidelines for the use of nonhuman laboratory primates and complied with the recommendations of the Weatherall report. The protocol of this study, monitoring parameters, and handling of the animals after the study were all approved by the Animal Welfare and Ethics Committee of IMBCAMS (Approval No. of ethics: DWSP2017050-6). Animals were humanely euthanized after reaching a humane endpoint.

### 2.2. Vaccine Preparation

A DTacP-sIPV quadrivalent vaccine was developed by IMBCAMS, and the preparation process is as follows: B. pertussis, Corynebacterium diphtheriae, and Clostridium tetanal were separately fermented. The components of pertussis antigens PT, FHA, and PRN were purified by column chromatography. The purified PT antigen was detoxified by aldehyde, and FHA was detoxified by formaldehyde. Diphtheria toxoid and tetanus toxoid were obtained from the fermentation harvest of Corynebacterium diphtheriae and Clostridium tetanal, respectively, by clarification, salting out, and detoxification by formaldehyde. The bulk of each component was obtained by adsorption with aluminum hydroxide adjuvant. The Sabin strain of poliovirus was inoculated into Vero cells cultured on microcarriers. The virus harvest was purified and inactivated by formaldehyde to obtain monovalent bulk. The formulated final bulk was proportionally prepared, filled, and packaged as a finished quadrivalent vaccine. Two experiment quadrivalent vaccines DTacP-sIPV were prepared using different formulae with varying contents of antigens (group 1 and group 2). The formulation of each 0.5 mL dose is shown in Table 1. Reference vaccines of groups 3, 4, and 5 were selected for comparison based on the DTacP-sIPV components. The DTacP-wIPV component of the PENTAXIM DTacP-wIPV/Hib pentavalent vaccine (TOA241M, Sanofi Pasteur, France) and the DTaP component of the DTaP/Hib quadrivalent vaccine (A20190615, Beijing Minhai Co., Beijing, China) were used for injection. The control wP vaccine used was purchased from the National Institutes for Food and Drug Control (NIFDC) of China; the negative control was set using aluminum hydroxide adjuvant without any vaccine components (group 6).

### 2.3. Grouping, Vaccination, and Pertussis Challenge

The schematic of the vaccination, challenge, and specimen collection timeline is displayed in Figure 1. Rhesus macaques were randomly assigned into five vaccine immunization groups (five in each group) and one blank control group (four macaques). A 0.25 mL dose of the corresponding vaccine was administered as a deep intramuscular injection into the lateral aspect of the thigh at 0, 1, and 2 months. Blood samples were obtained before and 30 days after each immunization, before aerosol challenge, and at days 3, 7, 14, 21, 28, 35, and 45 after challenge. The sera were immediately separated and stored at −20 °C for antibody and cytokines analysis. 

The rhesus macaques were challenged with B. pertussis (strain No.18323/CMCC58030; batch No. 2012003, obtained from NIFDC and cryopreserved by IMBCAMS) at day 28 after vaccination via aerosol exposure using an aerosolization apparatus designed by our lab and produced by Lanfang Honlan Equipment Co. (Lanfang Honlan, Lanfang, China) Each monkey was challenged with 105 CFU/mL for 60 min and separately housed as previously described [16].

### 2.4. Evaluation of Animal 

Each monkey cage was monitored by audio recording equipment. The number of coughs was monitored every day during 8:00–8:30, 12:00–12:30, and 17:00–17:30 until coughs disappeared. The number of coughs was determined for each 30-min observation period. The average number of coughs per hour for each day was calculated as the mean for all three observation periods. The rectal temperature in each group was measured before the aerosol challenge and at days 3, 7, 14, 21, 28, and 35 after the challenge to monitor temperature change. Blood samples (200 µL) were obtained in single-use sterile anticoagulated blood sampling tubes before aerosol challenge and at days 3, 7, 14, 21, 28, 35, and 45 after challenge for white blood cell (WBC) number determination by blood cell counting. The bacterial-positive nasopharyngeal wash (NPW) was collected at the same time, suspended in 1 mL of saline, and diluted 100- and 400-fold. Then, the 100 μL dilution was cultured in B–G medium (containing 30% sterile defibrinated sheep blood and 40 μg/mL cephalexin) and incubated at 37 °C. The number of colonies was determined after 5 days.

### 2.5. Serology Antibody 

Antibodies against PT, PRN, FHA, DT, and TT were detected using indirect enzyme-linked immunosorbent assays (ELISAs). The criteria for positive conversion rates of antibodies were as follows: PT-Ab ≥ 20 EU/mL, PRN-Ab ≥ 20 EU/mL, FHA-Ab ≥ 20 EU/mL, DT-Ab ≥ 0.1 IU/mL, TT-Ab ≥ 0.1 IU/mL. Antibody seroconversion for DTacP components was defined as a ≥ 4-fold increase from pre- to post levels. Neutralizing antibodies to poliovirus types I, II, and III were measured with a microneutralization test according to the method recommended by WHO [18]. An antibody titer ≥ 8 for each polio type was considered as a positive conversion, while a four-fold increase in the antibody level after immunization was considered as a positive conversion when tested positive for antibodies before immunization.

### 2.6. Measurement of Cytokines 

A total of six cytokines, including interferon (IFN)-γ, interleukin (IL)-13, IL-1β, IL-6, tumor necrosis factor (TNF)-α, and IL-12 (p40) were tested using Milliplex Cytokine Kits (Merck Millipore, Burlington, MA, USA). The fluorescence of cytokines was measured using a Luminex 200 system and analyzed by Milliplex Analyst software (version 5.1.0.0) using the 5-PL method. Seven measurements were obtained to construct a standard curve. An unpaired t-test was used to test for differences between the pre-challenge and peak cytokine production post-challenge period (3, 7, 14, 21, and 28 days after challenge) for each animal, owing to the high variability in starting concentrations between animals as well as the variability in the peak response for each cytokine post-infection. All sample measurements were repeated in duplicates.

### 2.7. Histopathological Observations of a Rhesus Macaque after Challenge

By day 11 post challenge, one rhesus macaque from the negative control group had been humanely euthanized because of the consistent decrease in the body temperature and severe diarrhea. The lung, trachea, and bronchi were fixed in formalin, sectioned, and stained to obtain tissue sections. Pathological features of tissue sections were observed under a microscope as previously described [16].

### 2.8. Statistics

At each blood sampling time point, the antibody seroconversion rates against all the antigenic components of the vaccine were computed with their 95% confidence intervals (CIs). Antibody geometric mean titers (GMTs)/geometric mean concentrations (GMCs) with 95% CIs were calculated by taking the log-transformed individual titers and calculating the anti-logs of the means of these transformed values. For polio antibodies, an individual neutralizing antibody titer lower than the detection limit (i.e., 1:4) was given an arbitrary value of 1:1, while for others, antibody concentrations below the cut-off of the assay were given an arbitrary value of half the cut-off. Antibody titers below the cut-off value for the assay were given an arbitrary value of half the cut-off value.

All data obtained in the experiments were processed and analyzed with the SPSS 18.0 software. For the data in accordance with a normal distribution, an ordinary ANOVA test was used to compare the difference, while for the data not in accordance with a normal distribution, the Kruskal–Wallis nonparametric test was used to compare the difference. A value of *p* < 0.05 was considered significant. Plots were constructed with Prism version 7.0 (GraphPad Software, Inc., San Diego, CA, USA).

## 3. Results

### 3.1. Antibody Response after Vaccine Immunization

The negative control group (group 6) showed no antibodies against all antigenic components both before and after immunization with all tested doses. The seroconversion rates of anti-PT for groups 1, 2, 3, and 4 reached 100% after the first vaccination dose, while the seroconversion rate for group 5 (wP group) reached 60% after three doses of immunization. The seroconversion rates of anti-PRN for groups 1, 2, 4, and 5 were 40%, 60%, 60%, and 0%, respectively, after one dose of immunization, and reached 100% after two doses. The conversion rates of anti-FHA were 100% after one dose for groups 1, 2, 3, and 4, and was 40% after one dose and 100% after two doses for group 5. As for anti-DT and anti-TT, the positive conversion rates for groups 1, 2, 3, and 4 reached 100% and were stably maintained after one dose. The conversion rate of IPV type I, II, and III antibodies for groups 1, 2, and 3 were 100% after three doses of immunization (Table 2).

The GMCs/GMTs against each antigen component of the combined vaccines before and after each dose of immunization was listed in Appendix A. As the antibody levels were compared, the anti-PT antibody titer for group 3 was higher than groups 1, 4, and 5 after the first doses; however, the difference was only investigated with groups 4 and 5 after the second and third doses. After the third dose, the GMC value of the anti-PT antibody for group 3 (1243.3 IU/mL) was significantly higher than that for group 4 (145.1 IU/mL) and group 5 (53.7 IU/mL) (*p* < 0.05). The GMC for groups 1 and 2 were 572.4 IU/mL and 811.2 IU/mL, respectively, and were not different other groups (Figure 2A). The GMC value of anti-FHA antibody in group 3 (914 IU/mL) was significantly higher than that reported for group 4 (192.5 IU/mL) and group 5 (180.3 IU/mL) (*p* < 0.05) but did not significantly differ from that observed for group 1 (605.9 IU/mL) and group 2 (613.5 IU/mL) after the third doses too (Figure 2B). For the GMC value of anti-PRN, no difference was observed among all the PRN component groups, groups 1, 2, 4, and 5 (Figure 2C). The anti-polio type 2 level was higher in group 3 (724.1 EU/mL) than in group 1 (16.0 EU/mL) and group 2 (4 EU/mL) (*p* < 0.05) after the first dose but did not differ after the second and third doses (Figure 2G). For type 1 and type 3 antibodies, all three groups showed the same antibody levels (Figure 2F,H).

### 3.2. Antibody Responses after Aerosol Challenge 

We challenged rhesus monkeys 30 days after the last vaccination dose. The level of anti-PT immediately increased in all five vaccination groups and reached its peak at 14, 28, 21, 21, and 28 days for groups 1, 2, 3, 4, and 5, respectively. On the contrary, the anti-PT level in group 6 (negative control) increased from 21 days, reached its peak at 28 days, and gradually decreased thereafter. The anti-PT level remained stable in other vaccination groups (Figure 3A). Furthermore, anti-PT level was higher on day 21 (*p* < 0.05) and day 28 (*p* < 0.01) than on day 3 in group 5 (wP). Similarly, the anti-PT level on day 28 was higher in group 5 than in group 1. The anti-FHA level significantly increased from day 3, peaked at day 14, and gradually decreased thereafter in group 6 (*p* < 0.01), while it tended to be steady after a slight increase from day 21 to 35 in the other five vaccination groups (Figure 3B). At day 14, the anti-FHA level in group 6 was significantly higher than that in group 4 (*p* < 0.05) and group 5 (*p* < 0.01). At day 14 after challenge, the anti-PRN level significantly increased in group 6 (*p* < 0.01) and showed an increasing trend in other groups and peaked at day 21 (group 1–4) and day 28 (group 5). The anti-PRN level slowly but stably decreased thereafter (Figure 3C).

### 3.3. Changes in Cytokine Levels of Rhesus Macaques after Challenge

The serum levels of IFN-γ, IL-13, IL-1β, IL-6, TNF-α, and IL-12 (p40) were tested at days 3, 7, and 14 post-challenge. Given the high variability in starting concentrations between animals and the variability in the peak response of each cytokine post-infection, an unpaired t-test was used to investigate differences between cytokine production during pre-challenge and post-challenge periods (3, 7, 14, 21, and 28 days after challenge) for each animal (Appendix A). Except the elevation in the TNF-α levels in group 5 between the pre- and post-challenge, which were significant (*p* < 0.05), only trends of elevation among groups were observed. INF-γ levels were elevated after the challenge in groups 1 and 3. IL-12 levels increased to varying degrees after the challenge in all groups, with more significant changes observed in groups 1, 3, and 4. The increase in TNF-α levels was maximum in group 3, 4, and 5, while the increase in IL-1β levels was observed in all except group 2. Considering Th2-associated cytokines, IL-6 levels were elevated in groups 2, 3, 5, and 6 and IL-13 levels were elevated in groups 1, 4, 5, and 6.

### 3.4. DTacP-sIPV Protection Efficacy against Pertussis

To investigate the efficacy of the vaccine developed against the clinical symptoms of pertussis, the number of coughs of each rhesus macaque was counted after the viral challenge. The animals from the negative control group (group 6) developed an obvious cough at day 2 that peaked at day 4 (102 times per hour) and then slowly declined and disappeared at day 14 following the challenge. For group 3 animals, cough symptoms were observed from day 2 and peaked at day 4 (55 times per hour) and were relieved rapidly thereafter. The rhesus macaques from other vaccine groups did not cough during the entire investigation period (Figure 4A).

The dynamic profiles of leukocytosis were also determined. There was an obvious increase in the WBC count in negative control group (group 6) macaques at day 3. The WBC count peaked at day 7 (4.2 × 10^10^/L) and gradually declined thereafter to the normal level. The WBC counts for other vaccinated groups remained steady and showed no significant changes (*p* > 0.05) (Figure 4B).

We observed nasopharyngeal colonization of B. pertussis in the animals from the negative control group (group 6) at day 3 following the challenge that slowly decreased but was not cleared within 45 days. For groups 1, 2, 3, and 4 animals, B. pertussis was identified at day 3 after challenge, peaked at day 7, and rapidly decreased and cleared up YEby day 45. The number of nasopharyngeal colonies was the least for the wP vaccination group (group 5), which showed a peak at day 7 followed by a rapid decrease in the count; B. pertussis was cleared by day 28. The clearance time and peak value of colony counts were significantly lower in all the vaccination groups than in the negative control group (*p* < 0.05). In addition, these parameters were significantly lower in the wP group than in the other vaccination groups (*p* < 0.05) (Figure 4C).

We measured the rectal temperature after the challenge and found no significant differences between all groups.

### 3.5. Pathological Characteristics of the Rhesus Macaque from the Negative Control Group after Challenge

By day 11 post-challenge, one rhesus macaque from the negative control group was humanely euthanized after reaching the humane endpoint. Its WBC count increased to 7.509 × 10^10^/mL at day 7 after the challenge, and 107.88 CFU/mL of B. pertussis was found in the lung tissue at the time of death. The lung and trachea of the dead rhesus macaque were stained for histological examination. The upper lobe of the left lung showed consolidation, protein exudate in the pulmonary alveolar spaces, thickening of the alveolar septum, infiltration of inflammatory cells, destruction of the bronchial epithelium, and vascular congestion (Figure 5A). The lower lobe of the left lung showed alveolar septum hemorrhage with hemosiderosis, consolidation, exudation of alveolar proteins, and inflammatory cells (Figure 5B). In addition, destruction of the bronchial epithelium and bronchioles with local suppurative inflammation was observed in the middle lobe of the right lung (Figure 5C), and mucosal destruction of the bronchial epithelium and bronchioles with acute infiltration of inflammatory cells, and alveolar protein exudate were reported in the lower lobe of the right lung.

## 4. Discussion

The successful development of sIPV in China in 2015 served as a starting point for the establishment of DTaP-sIPV combined vaccines to improve vaccination compliance and coverage rates through inclusion in routine immunization. In the present study, the novel DTacP-sIPV combined vaccine was developed and its immunogenicity and protective efficacy were investigated in a rhesus model following comparison with commercialized pertussis combined vaccines. 

After the first DTaP vaccine developed in Japan in 1984, several aP vaccines have been developed and used worldwide as a substitute to DTwP in routine immunization [19]. The DTacP-wIPV/Hib pentavalent vaccine by Sanofi Pasteur exhibited a good immunogenic effect with >97%, >86%, and >88% seroprotection/seroresponse rates for each component in clinical trials in the US, Canada, and China, respectively [3,20]. Meanwhile, the DTaP/Hib quadrivalent vaccine by Beijing Minhai has been proved to be effective [21]. In the present study, the antibodies against PT, FHA, PRN, DT, TT, and three types of polios were tested after the immunization of rhesus macaques with three doses at 0, 1, and 2 months. The seroconversion rate of each component reached 100% after three doses of primary immunization in all tested vaccination groups, although the levels of PT, FHA, PRN, and polio antigens were different in two DTacP-sIPV groups and no significant differences were observed. The GMT/GMC induced in the vaccine groups exhibited some differences for anti-PT, anti-FHA, and anti-PRN during the vaccination period. However, after the third dose, only DTacP-wIPV/Hib vaccinated monkeys showed higher anti-PT and anti-PHA than DTaP/Hib- and wP-vaccinated monkeys. For polio antibodies, the anti-type 2 antibody titer was higher in group 3 than in groups 1 and 2 after the first dose of vaccination; however, no significant difference was observed after the second and third doses. Thus, formulations 1 and 2 DTacP-sIPV showed good immunogenicity, as evident from no difference from the reference DTacP-wIPV/Hib, DTaP/Hib, and wP vaccines or no difference between the two polio antigen formulations. 

The protective efficacy of aP is considered to be not as good as that of the wP vaccine [22,23]. Thus, we challenged infant rhesus macaques and compared the protective efficacy of DTacP-sIPV with that of the reference commercial combined vaccine and wP. After the challenge, the negative control group monkeys showed clinical symptoms similar to those previously reported in a nonhuman private pertussis infection model [16,24]. The number of circulating WBCs in the peripheral blood significantly increased at day 7; the test animals developed a cough on the first day that peaked to 102 coughs per hour at day 4, then gradually decreased and stopped 15 days later; B. pertussis colonization in the nasopharyngeal cavity immediately peaked at day 3 and gradually decreased but still remained until 45 days. On the contrary, no significant change in cough frequency was observed in all animals from the vaccine groups, except those from group 3 (DTacP-wIPV/Hib). However, the number of coughs in group 3 was lower than that in the negative control group and gradually disappeared after day 5. Previous studies in a baboon model suggested the inability of the aP vaccine to prevent the colonization and clearance of infection as compared with the wP vaccine [15]. In the present study, the nasopharyngeal colonization of B. pertussis detected at day 3 peaked at day 7 after the challenge in all groups; however, the colonies in the wP vaccinated group rapidly decreased and cleared up by day 28; all aP groups showed a gradual decrease in the number of colonies at day 45. These results suggest that though all the vaccination groups could effectively prevent the colonization of pertussis, the number of pertussis colonies in the wP group was significantly lower than that in the aP-related groups and that the clearance of pertussis infection was significantly earlier than that in the aP and negative control groups. Thus, wP was better at preventing pertussis infection than the aP vaccine [23,25].

The aP vaccine used in this study could prevent the development of clinical symptoms of pertussis but was unable to prevent pertussis infection effectively as the wP vaccine. Follow-up studies on new pertussis vaccines should be conducted to determine a more effective vaccine against pertussis.

## 5. Conclusions

In the present study, antibody-positive conversion rates, antibody levels, and cytokine levels showed that the two formulation candidates DTacP-sIPV quadrivalent vaccine could induce a good immune response in rhesus macaques; no significant difference was observed as compared to the control vaccines. The clinical symptoms did not significantly differ between the rhesus macaques from different vaccination groups after the challenge and were superior to those observed in the negative control group. Thus, the candidate vaccine could prevent the development of clinical symptoms of pertussis and had good efficacy and may be used for new DTacP-sIPV development.

## Figures and Tables

**Figure 1 vaccines-10-00047-f001:**
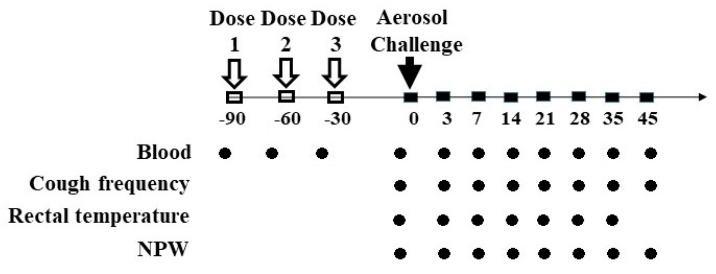
Timeline of vaccination, B. pertussis aerosol challenge, and sample collection from rhesus macaques. Infant rhesus macaques were vaccinated with different vaccines at −90, −60, and −30 days with 0-, 1-, and 2-month schedules and were infected with B. pertussis by aerosol challenge on day 0 (hollow arrow). Blood samples were collected before and 30 days after each immunization, before aerosol challenge, and at days 3, 7, 14, 21, 28, 35, and 45 after challenge. The cough frequency, WBC count, and NPW were determined before aerosol challenge and at days 3, 7, 14, 21, 28, 35, and 45 after challenge. The rectal temperature in each group was measured before aerosol challenge and at days 3, 7, 14, 21, 28, and 35 after challenge.

**Figure 2 vaccines-10-00047-f002:**
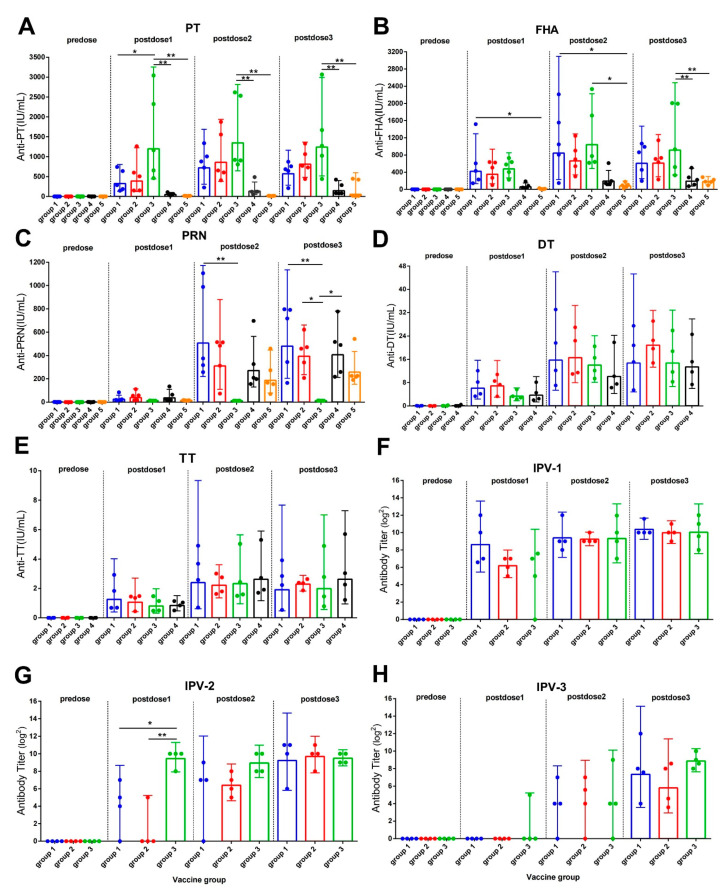
Antibody levels in the rhesus macaques from different groups after immunization with vaccines. (**A**–**E**): Anti-PT, anti-PRN, anti-FHA, anti-DT, and anti-TT antibodies tested using ELISA. Standard serum was prepared using International Standard for Pertussis Antiserum (Human Resource) (NIBSC Code: 06/140, provided by NIFDC), National Standard for Tetanus Human Immunoglobulin (Batch No. 001, Specification: 10 U/vial, provided by NIFDC), and International Standard for Diphtheria Human Antibody (Batch No. 10/262, Specification: 2 U/vial, provided by NIBSC). (**F**–**H)**: Type I, II, and III polio antibodies tested using neutralization method with Hep-2 cells. The results of the negative control (group 6) were below the detection limit and hence not shown in the figure. Data were analyzed using one-way ANOVA with Tukey’s multiple comparison test. * *p* < 0.05, ** *p* < 0.01 (*n* = 4).

**Figure 3 vaccines-10-00047-f003:**
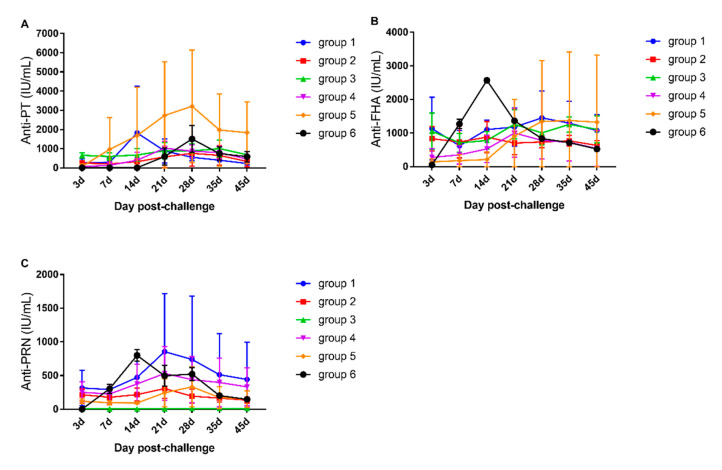
Antibody responses after B. pertussis aerosol challenge in immunized rhesus macaques from different groups. (**A**–**C**): Anti-PT, anti-PRN, and anti-FHA antibody levels were tested using ELISA. Standard serum was prepared using International Standard for pertussis (Human resource) (NIBSC code: 06/140, from NIFDC)

**Figure 4 vaccines-10-00047-f004:**
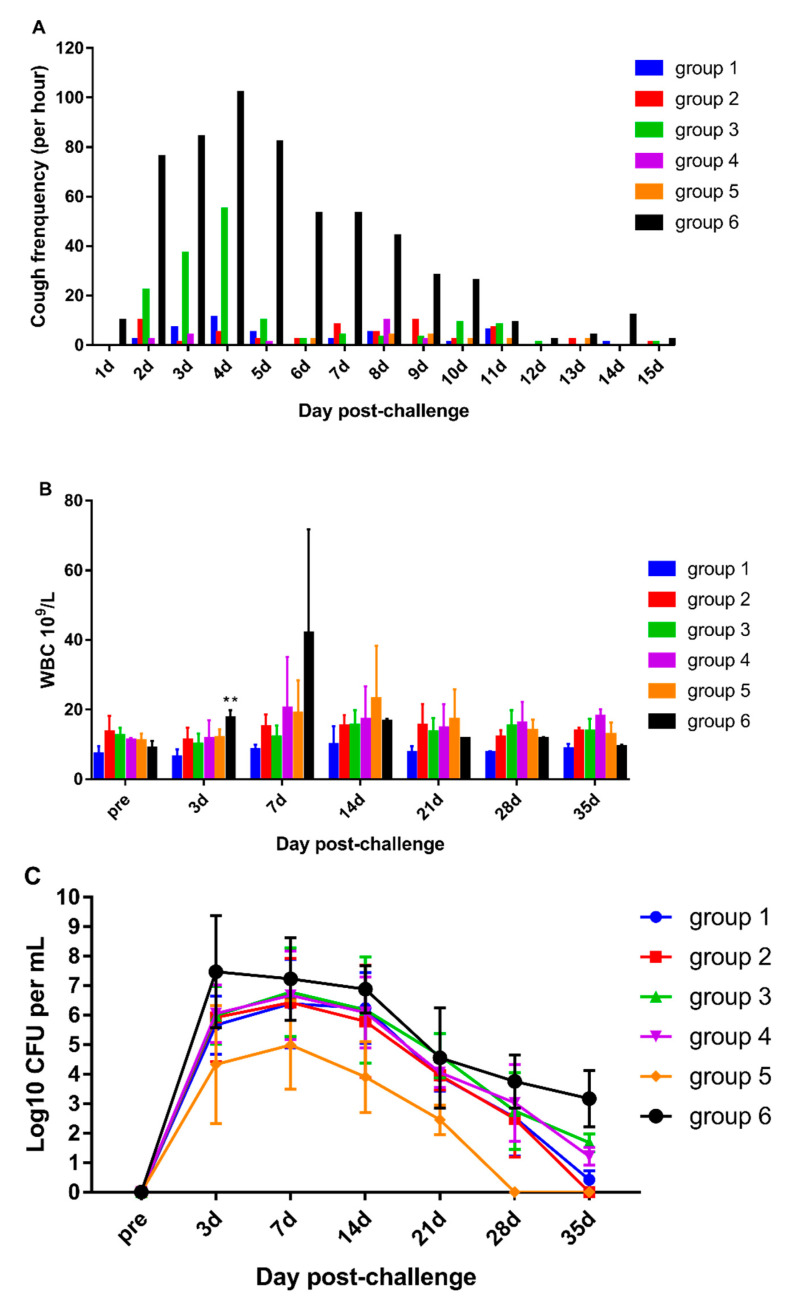
Cough count, leukocytosis, and B. pertussis colonization in rhesus macaques from different groups after B. Pertussis aerosol challenge. (**A**): Number of coughs after challenge, as recorded by audio equipment and calculated during three time periods each day (8:00–8:30, 12:00–12:30, 17:00–17:30). (**B**): White blood cell count of rhesus macaques after challenge. The number of circulating white blood cells before and after challenge was determined and compared by a blood cell counter, ** *p* < 0.01. (**C**): Colonization of the nasopharynx of rhesus macaques after challenge. Nasopharyngeal swabs were collected, and the recovered swabs were plated on B–G plates for 4 days for colonization.

**Figure 5 vaccines-10-00047-f005:**
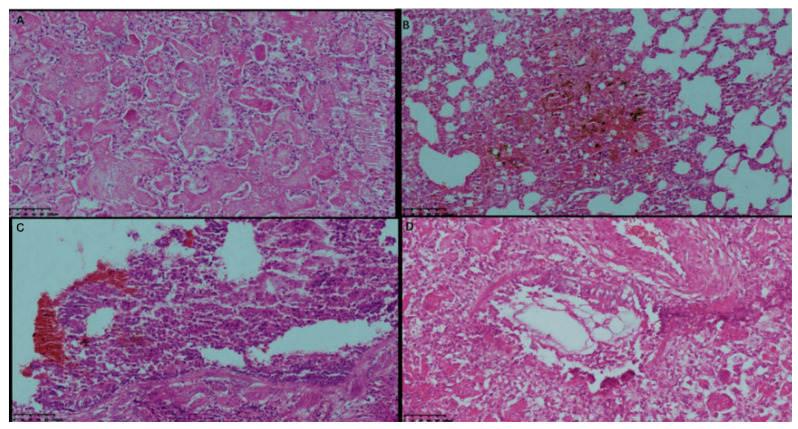
Pathological sections of the dead rhesus macaque after pertussis challenge. (**A**): Thickening of the alveolar septum and infiltration of inflammatory cells in the upper lobe of the left lung. (**B**): Lung consolidation, exudation of alveolar proteins, inflammatory cells, and alveolar septum hemorrhage in the lower lobe of the left lung. (**C**): Destruction of the bronchial epithelium and bronchioles with local suppurative inflammation in the middle lobe of the right lung. (**D**): Mucosal destruction of the bronchial epithelium and bronchioles in the lower lobe of the right lung.

**Table 1 vaccines-10-00047-t001:** Composition and grouping of experimental and control vaccines used in the study.

Vaccine Component	Group1 (DTacP-sIPV Formulation1)	Group2 (DTacP-sIPV Formulation2)	Group3 (DTacP-wIPV/Hib)	Group4 (DTaP/Hib)	Group5 (wP)	Group6 (Negative Control)
DT (Lf/dose)	25	12.5	15	12.5	—	—
TT (Lf/dose)	10	3.5	5	3.5	—	—
PT (μg/dose)	25	25	25	9 µg PN *	—	—
PRN (μg/dose)	8	8	—		—	—
FHA (μg/dose)	25	25	25		—	—
Whole cell pertussis (IU/dose)	—	—	—	—	4	—
IPV type I (DU/dose)	30 (Sabin)	15 (Sabin)	40 (Mahoney)	—	—	—
IPV type II (DU/dose)	32 (Sabin)	16 (Sabin)	8 (MEF-1)	—	—	—
IPV type III (DU/dose)	45 (Pfizer)	22.5 (Pfizer)	32 (Saukett)	—	—	—
Aluminum (mg/dose)	0.60–0.75	0.60–0.75	0.60–0.75	0.60–0.75	—	0.60–0.75

Remark: —, does not contain this component; * PN, the total protein nitrogen of PT, PRN and FHA manufactured by co-purified process.

**Table 2 vaccines-10-00047-t002:** Seroconversion rates (%) against different components of the combined vaccines after each dose of immunization.

	Group 1			Group 2			Group 3			Group 4			Group 5			Group 6		
	Dose 1	Dose 2	Dose 3	Dose 1	Dose 2	Dose 3	Dose 1	Dose 2	Dose 3	Dose 1	Dose 2	Dose 3	Dose 1	Dose 2	Dose 3	Dose 1	Dose 2	Dose 3
PT	100%	100%	100%	100%	100%	100%	100%	100%	100%	100%	100%	100%	0%	0%	60%	-	-	-
FHA	100%	100%	100%	100%	100%	100%	100%	100%	100%	100%	100%	100%	40%	100%	100%	-	-	-
PRN	40%	100%	100%	60%	100%	100%	-	-	-	60%	100%	100%	0%	100%	100%	-	-	-
DT	100%	100%	100%	100%	100%	100%	100%	100%	100%	100%	100%	100%	-	-	-	-	-	-
TT	100%	100%	100%	100%	100%	100%	100%	100%	100%	100%	100%	100%	-	-	-	-	-	-
Polio I	100%	100%	100%	100%	100%	100%	75%	100%	100%	-	-	-	-	-	-	-	-	-
Polio II	75%	75%	100%	25%	100%	100%	100%	100%	100%	-	-	-	-	-	-	-	-	-
Polio III	0%	75%	100%	0%	75%	100%	25%	75%	100%	-	-	-	-	-	-	-	-	-

Remark: “-”, does not contain this component. The criteria for positive conversion rates of antibodies were as follows: PT-Ab ≥ 20 EU/mL, PRN-Ab ≥ 20 EU/mL, FHA-Ab ≥ 20 EU/mL, D-Ab ≥ 0.1 IU/mL, T-Ab ≥ 0.1 IU/mL. IPV-I-Ab, IPV-II-Ab, and IPV-III-Ab titer ≥ 8 were determined as positive conversion. If the antibody was positive before immunization, the four-fold increasing of the antibody level after immunization was considered as positive conversion.

## Data Availability

The data presented in this study are available on request from the corresponding author. The data are not publicly available due to the new vaccine development secrecy.

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
