# Peer review of "Immunogenicity of a Candidate DTacP-sIPV Combined Vaccine and Its Protection Efficacy against Pertussis in a Rhesus Macaque Model"

_vaccines, 2021, doi:10.3390/vaccines10010047_

Round 1
Reviewer 1 Report
Introduction
The introduction is concise and motivates the comparisons undertaken.
Methods
This is a well planned comparison of five vaccine formulations compared to a control group.
Please comment on how well macaques match human children as vaccine candidates.
Recording coughs for only 1/2 an hour three times daily initially seemed rather minimal but some animals were coughing > 100 x/hour.
Bacterial-positive nasopharyngeal washes were a second important clinical finding.
How was the choice of challenge strain made?
The vaccine was administered at 0, 1 and 2 months. How do these intervals compare to the intervals in most countries?
Blood samples were collected before and 30 days after vaccination and then on a frequent basis after challenge: 3, 7, 14, 21, 28, 35 and 45 days.
Results
Table 2 shows high seroconversion rates.
Thus there are extensive data per macaque. However, the issue of the statistical validity of the results with 5 macaques/group is not addressed.
For example, anti-PT Ab levels in group 3 were 1495 +/- 984 IU/ml and in group 5 183 +/- 231 IU/ml. What is the generalisability of these results to human populations in which millions of children with varying immune systems and nutrition levels will be vaccinated?
Author Response
Reviewer 1
We thank the Reviewer’s for carefully reading the manuscript and providing insightful comments. We have revised the manuscript to address all comments point-by-point as below:
Introduction
The introduction is concise and motivates the comparisons undertaken.
Methods
This is a well planned comparison of five vaccine formulations compared to a control group.
Please comment on how well macaques match human children as vaccine candidates.
Response: Thanks for your comment! As one of the non-human primates, rhesus macaques has been selected as an important animal model for human infectious disease as well as for vaccine evaluation. Our previous pertussis animal model suggested that upon aerosol infection, the infant rhesus macaques infected with the recent clinically isolated B.pertussis developed typical whooping cough, leukocytosis, bacteria-positive nasopharyngeal wash, and inter-animal transmission, as same as the human children did. However it was not investigated in the adult monkey. In present study, we selected infant macaques (aged 3-4 months) to evaluate the candidate vaccine efficacy. We added “According to our previous successful pertussis model on infant rhesus macaques pertussis challenge model, showing typical whooping cough, leukocytosis, bacteria-positive nasopharyngeal wash, inter-animal transmission, and the systemic immune response after pertussis infection, which matched human children infected with pertussis” in the method section (Line 83-86).
Recording coughs for only 1/2 an hour three times daily initially seemed rather minimal but some animals were coughing > 100 x/hour.
Response: Thanks for your comment! In the baboo model and our infant rhesus model experiments, the numbers of coughs were recorded four times daily, while it was three times daily in this study [1,2]. In our study with infant monkeys model, the average number of coughs for recording three times and four times did not show any differences. In addition, as there were a total of 29 monkeys to be monitored, we recorded three times daily for this study. Even in the negative control group with maximal numbers of cough, the coughing> 100/hour were only investigated at day 4, and normally ranged from 2- 84 during 15 days. We added “The number of coughs was determined for each 30-minute observation period. The average number of coughs per hour for each day was calculated as the mean for all three observation periods. ” in the revised manuscript to make the cough calculating method clear (Line 150-152 ).
Bacterial-positive nasopharyngeal washes were a second important clinical finding.
How was the choice of challenge strain made?
Response: In present study, we selected B.p. 18323 as the challenge strain. The 18323 strain was identified as one of the most virulent B.p strains in the 1940s and used as an intra-cerebral challenge strain in most studies worldwide [3]. In China, the National Institutes for Food and Drug Control authorized B.p 18323 as a challenge strain for potency study for all the commercial pertussis vaccines. Thus, we selected it to evaluate the DTacP-sIPV efficacy of pertussis for the future application for clinical trials.
The vaccine was administered at 0, 1 and 2 months. How do these intervals compare to the intervals in most countries?
Response: Thanks for your comment! We chose 0, 1 and 2 months immunization procedure was because of the routine immunization schedule of DTaP vaccine and sIPV vaccine in China, which are 3-dose schedule with 4 weeks intervals, and the DTaP based combined vaccine such as DTaP-IPV-Hib and DTaP-Hib in private market were also using 3 doses schedule with 4 weeks intervals too in China and many other countries worldwide [4,5]. For DTacP based combined vaccine, there were totally three schedules for primary immunization at present worldwide: three vaccine doses administered at age 2, 4, and 6 months as standard schedule in most countries like in Latin America and Asia; 2, 3, and 4 months as ‘accelerated’ schedule in Europe; 6, 10, and 14 weeks as expanded program of immunization schedule in South Africa and India. However, no significant difference was investigated among these schedules [6].
Blood samples were collected before and 30 days after vaccination and then on a frequent basis after challenge: 3, 7, 14, 21, 28, 35 and 45 days.
Response: Thanks for your reminder! We collected the blood samples before and 30 days after each vaccination (1, 2 and 3 doses), and at day 3, 7, 14, 21, 28, 35, and 45 after challenge. All of the samples were detected for antibodies, while the samples of pre-challenge and post-challenge periods (3, 7, 14, 21, 28 days after challenge) were detected for cytokines. We have added it in the revised manuscript (Line 132)
Results
Table 2 shows high seroconversion rates.
Thus there are extensive data per macaque. However, the issue of the statistical validity of the results with 5 macaques/group is not addressed.
For example, anti-PT Ab levels in group 3 were 1495 +/- 984 IU/ml and in group 5 183 +/- 231 IU/ml. What is the generalisability of these results to human populations in which millions of children with varying immune systems and nutrition levels will be vaccinated?
Response: Thanks for your comment! We added supplementary table1 listed the GMCs/GMTs against different antigens of the combined vaccines after each dose of immunization. GMTs /GMCs with 95% CIs were calculated by taking the log-transformed individual titers and calculating the anti-logs of the means of these transformed values. For polio antibodies, an individual neutralizing antibody titer lower than the detection limit (i.e., 1:4) was given an arbitrary value of 1:1, while for others, antibody concentrations below the cut-off of the assay were given an arbitrary value of half the cut-off. We added the statistics method in the method section (Line 194-197) and revised the result of antibody level comparing using GMC data (Line 218-233 ).
Just as you indicated, the antibody levels among monkeys exhibited large ranges in present study. This may caused by limited number of animals per group (5 in vaccination groups, 4 in negative control group) and individual difference which have been investigated in monkey study and baboo studies [7,8]. During statistics analysis, for the data in accordance with normal distribution, ordinary ANOVA test was used to compare the difference, while for the data not in accordance with normal distribution, Kruskal-Wallis nonparametric test was used to compare the difference(Line 200-202 ). The present study focused on the analysis of the antibody response induced by the two formulations of DTacP-sIPV vaccines comparing with the commercial vaccines DTacP-wIPV and DTaP/Hib, which suggested that the present DTacP-sIPV could be a candidate vaccine for the future clinical trials, however the induvial response difference should be analyzed based on the real children’s data.
Reference
- Warfel, J.M.; Zimmerman, L.I.; Merkel, T.J. Acellular pertussis vaccines protect against disease but fail to prevent infection and transmission in a nonhuman primate model. Proc Natl Acad Sci U S A 2014, 111, 787-792, doi:10.1073/pnas.1314688110.
- Mingbo, S.; Yan, M.; Yinhua, X.; Huijuan, Y.; Li, S.; Yanchun, C.; Guoyang, L.; Shude, J.; Shumin, Z.; Qihan, L. Dynamic profiles of neutralizing antibody responses elicited in rhesus monkeys immunized with a combined tetravalent DTaP-Sabin IPV candidate vaccine. Vaccine 2014, 32, 1100-1106.
- Xing, D.; Das, R.G.; O'Neill, T.; Corbel, M.; Dellepiane, N.; Milstien, J. Laboratory testing of whole cell pertussis vaccine: a WHO proficiency study using the Kendrick test. Vaccine 2001, 20, 342-351, doi:10.1016/s0264-410x(01)00372-3.
- Capeding, M.R.; Cadorna-Carlos, J.; Book-Montellano, M.; Ortiz, E. Immunogenicity and safety of a DTaP-IPV//PRP approximately T combination vaccine given with hepatitis B vaccine: a randomized open-label trial. Bull World Health Organ 2008, 86, 443-451, doi:10.2471/blt.07.042143.
- Kummeling, I.; Thijs, C.; Stelma, F.; Huber, M.; van den Brandt, P.A.; Dagnelie, P.C. Diphtheria, pertussis, poliomyelitis, tetanus, and Haemophilus influenzae type b vaccinations and risk of eczema and recurrent wheeze in the first year of life: the KOALA Birth Cohort Study. Pediatrics 2007, 119, e367-373, doi:10.1542/peds.2006-1479.
- Syed, Y.Y. DTaP-IPV-HepB-Hib Vaccine (Hexyon((R))): An Updated Review of its Use in Primary and Booster Vaccination. Paediatr Drugs 2019, 21, 397-408, doi:10.1007/s40272-019-00353-7.
- Gurung, S.; Preno, A.N.; Dubaut, J.P.; Nadeau, H.; Hyatt, K.; Reuter, N.; Nehete, B.; Wolf, R.F.; Nehete, P.; Dittmer, D.P.; et al. Translational Model of Zika Virus Disease in Baboons. J Virol 2018, 92, doi:10.1128/JVI.00186-18.
- Jiang, W.; Wei, C.; Mou, D.; Zuo, W.; Liang, J.; Ma, X.; Wang, L.; Gao, N.; Gu, Q.; Luo, P.; et al. Infant rhesus macaques as a non-human primate model of Bordetella pertussis infection. BMC Infect Dis 2021, 21, 407, doi:10.1186/s12879-021-06090-y.
Reviewer 2 Report
The authors evaluate the candidate vaccine DTacp-sIPV and compare its efficacy to other vaccines formulation (DTap-wIPV/Hib, DTap/Hib and wP) in a rhesus macaque model. They confirm with pathological characterization of an infected animal that this animal model is relevant for testing the pertussis vaccines. The manuscript is clearly written and the experiments are well described.
For poliovirus vaccination, DTacp-sIPV has lower efficacy for IPV-2 since only third dose provide complete seroconversion. This is not clearly stated in line 343 of discussion: “no difference was observed after the second and third dose of vaccination.” But in Figure 2, group 1 has still low antibody titer after second dose.
The name of the formulation of group 3 is either stated DTacP-wIPV (line 106 and 329) or DTaP-wIPV is the rest of the text. Could the authors state the reason of difference
There is some misspelling
Line 107 and 329. “Sinofi” should be written “Sanofi”
Line 48 “poliovirus stain-based”
The alpha of TNF-alpha is sometimes missing
Line 200, the sentence “As the antibody level by compared…” is not clear
Author Response
We thank the Reviewer’s for carefully reading the manuscript and providing insightful comments. We have revised the manuscript to address all comments point-by-point as below:
The authors evaluate the candidate vaccine DTacp-sIPV and compare its efficacy to other vaccines formulation (DTap-wIPV/Hib, DTap/Hib and wP) in a rhesus macaque model. They confirm with pathological characterization of an infected animal that this animal model is relevant for testing the pertussis vaccines. The manuscript is clearly written and the experiments are well described.
For poliovirus vaccination, DTacp-sIPV has lower efficacy for IPV-2 since only third dose provide complete seroconversion. This is not clearly stated in line 343 of discussion: “no difference was observed after the second and third dose of vaccination.” But in Figure 2, group 1 has still low antibody titer after second dose.
Response: Thanks for your comment! The formulation of IPV in DTacP-sIPV was in accord with the commercial sIPV, the inactivated poliovirus vaccine made from attenuated poliovirus Sabin strain. Our previous clinical study found that the GMTs of anti-type 2 polio induced by sIPV was lower than that induced by OPV (oral poliovirus vaccine made by live attenuated poliovirus Sabin strain), and the sIPV induced lower immunogenicity for type 2 compared to type1 and type 3 [1]. Other studies also suggested the Sabin-IPV type 2 is less immunogenic than regular wild type IPV type 2 [2,3]. This may be related to the structure of the antigenic sites of the viral capsid protein and to the harmful effect of formalin treatment. However the immunogenicity was found non-inferior to that induced by wIPV control group after complete immunization [1,4,5], which was same as our results. Though the seroconversion rates were 75% and 25% for group 1 and group 2 after the first dose, respectively, it reached 100% and 75% with no significant difference compared to the DTap-wIPV/Hib control groups after the second dose. In addition, it reached 100% after the third dose in all groups. For GMTs, the difference was significant after 1 dose, but was not statistical significant after 2 and 3 doses. Thus, in the discussion, we stated that no significant difference was observed after the second and third dose for both the seroconversion rate and statistically significant difference was only found in GMTs after the first dose.
The name of the formulation of group 3 is either stated DTacP-wIPV (line 106 and 329) or DTaP-wIPV is the rest of the text. Could the authors state the reason of difference
Response: Thanks for your comment and sorry for our careless! DTaP-wIPV in the manuscript should be revised to DTacP-wIPV. The DTaP used co-purification process for pertussis, while DTacP used the purification process for the individual component of pertussis to improve the quality of the vaccine. In present study, we manufactured a three-component aP for DTacP-sIPV, and used DTacP-wIPV/Hib (two-component aP) and DTaP/Hib (co-purification process for pertussis) as control. We have corrected all the mistakes in the revised manuscript. Thanks again for your reminder!
There is some misspellish
Line 107 and 329. “Sinofi” should be written “Sanofi”
Line 48 “poliovirus stain-based”
The alpha of TNF-alpha is sometimes missing
Line 200, the sentence “As the antibody level by compared…” is not clear
Response: Thanks for your corrections and sorry for our careless! We have revised all the misspellings in the revised manuscript according to your comments!
Reference
- Liao, G.; Li, R.; Li, C.; Sun, M.; Li, Y.; Chu, J.; Jiang, S.; Li, Q. Safety and immunogenicity of inactivated poliovirus vaccine made from Sabin strains: a phase II, randomized, positive-controlled trial. J Infect Dis 2012, 205, 237-243, doi:10.1093/infdis/jir723.
- Westdijk, J.; Brugmans, D.; Martin, J.; van't Oever, A.; Bakker, W.A.; Levels, L.; Kersten, G. Characterization and standardization of Sabin based inactivated polio vaccine: proposal for a new antigen unit for inactivated polio vaccines. Vaccine 2011, 29, 3390-3397, doi:10.1016/j.vaccine.2011.02.085.
- Kersten, G.; Hazendonk, T.; Beuvery, C. Antigenic and immunogenic properties of inactivated polio vaccine made from Sabin strains. Vaccine 1999, 17, 2059-2066, doi:10.1016/s0264-410x(98)00409-5.
- Liao, G.; Li, R.; Li, C.; Sun, M.; Jiang, S.; Li, Y.; Mo, Z.; Xia, J.; Xie, Z.; Che, Y.; et al. Phase 3 Trial of a Sabin Strain-Based Inactivated Poliovirus Vaccine. J Infect Dis 2016, 214, 1728-1734, doi:10.1093/infdis/jiw433.
- Sun, M.; Li, C.; Xu, W.; Liao, G.; Li, R.; Zhou, J.; Li, Y.; Cai, W.; Yan, D.; Che, Y.; et al. Immune Serum From Sabin Inactivated Poliovirus Vaccine Immunization Neutralizes Multiple Individual Wild and Vaccine-Derived Polioviruses. Clin Infect Dis 2017, 64, 1317-1325, doi:10.1093/cid/cix110.